# Instrumental music training relates to intensity assessment but not emotional prosody recognition in Mandarin

**Mengting Liu**[1], **Xiangbin Teng**[2], **Jun Jiang**[3]*

**1** Department of Art, Harbin Conservatory of Music, Harbin, China, **2** Department of Psychology, The Chinese University of Hong Kong, Shatin, Hong Kong SAR, China, **3** Music College, Shanghai Normal University, Shanghai, China

* jjxytc@163.com

**Data Availability Statement:** The raw stimuli, procedure, data, and code are available from the Harvard Dataverse Repository at https://doi.org/10.7910/DVN/EEI1QR.

## Abstract

Building on research demonstrating the benefits of music training for emotional prosody recognition in nontonal languages, this study delves into its unexplored influence on tonal languages. In tonal languages, the acoustic similarity between lexical tones and music, along with the dual role of pitch in conveying lexical and affective meanings, create a unique interplay. We evaluated 72 participants, half of whom had extensive instrumental music training, with the other half serving as demographically matched controls. All participants completed an online test consisting of 210 Chinese pseudosentences, each designed to express one of five emotions: happiness, sadness, fear, anger, or neutrality. Our robust statistical analyses, which included effect size estimates and Bayesian factors, revealed that music and nonmusic groups exhibit similar abilities in identifying the emotional prosody of various emotions. However, the music group attributed higher intensity ratings to emotional prosodies of happiness, fear, and anger compared to the nonmusic group. These findings suggest that while instrumental music training is not related to emotional prosody recognition, it does appear to be related to perceived emotional intensity. This dissociation between emotion recognition and intensity evaluation adds a new piece to the puzzle of the complex relationship between music training and emotion perception in tonal languages.

## Introduction

Accurate recognition of emotional prosody is crucial for effective social interactions [1, 2]. Individuals adept at identifying others' emotional tones are better prepared for nuanced social engagement [3–6]. In contrast, those with challenges in understanding emotional prosody often face social complications [e.g., 7–12]. It is crucial to examine how individuals from varied linguistic backgrounds perceive emotional prosody. Exploring this, particularly the impact of specific training on social well-being through improved prosody recognition, unveils insights into cross-cultural variations in the utilization of acoustic cues, especially among individuals with and without tonal language experience [13, 14].

**Funding:** The author(s) received no specific funding for this work.

**Competing interests:** The authors have declared that no competing interests exist.

Recognizing emotional prosody hinges on acoustic features like pitch, intensity, tempo, and voice quality/timbre [15–17], which also serve as essential elements in music, such as Western tonal music. Different emotional prosodies are characterized by distinct features: for instance, happy prosody involves moderately high average pitch, high average voice intensity, and a rapid speech rate, whereas sad prosody is marked by low average pitch, low average voice intensity, and a slow speech rate [18–21]. Effective recognition of emotional prosody requires the efficient extraction of these acoustic markers and appropriate perceptual analysis. Given the shared acoustic elements between music and speech in conveying emotion [18, 22–24], instrumental music training could enhance speech perception [25], as posited by Patel's OPERA hypothesis [26, 27]. The OPERA hypothesis posits that shared sensory and cognitive mechanisms underlie both music and speech perception [28, 29], with music imposing more demanding conditions on these systems. Therefore, training in one domain (e.g., music listening) could have cross-modal benefits for the other (e.g., prosody recognition).

Furthermore, individuals with musical training and tonal languages listeners may exhibit advantages in perceiving lexical tones and musical pitch by paying greater attention to the relevant sound cues acquired in speech or music [13, 29]. Tonal languages listeners and musicians have an advantage in recognizing and learning lexical tones, exhibiting cross-domain transfer in pitch perception [13, 30]. Pitch plays a crucial role in both speech and music, serving as a predictive factor in the perception of emotional aspects of vocals. In nontonal languages, it is primarily utilized for emotional expression in speech. In contrast, tonal languages use pitch variations not only as emotional markers but also to convey semantic information [31–36]. As a result, the task of recognizing emotional prosody in tonal languages becomes more intricate, potentially requiring additional cognitive resources to mitigate interference from semantic aspects during emotion recognition. While music training is associated with enhanced perception of lexical tones in native speakers of tonal languages [37], it remains uncertain whether it is associated with improved perception of emotional prosody in their native language.

Many studies have examined the relationship between music training and emotional prosody recognition in native speakers of nontonal languages, such as English, German, Portuguese, Dutch, and Hebrew [38–41] (see [42–44], for reviews). Most of these studies utilize a cross-sectional design comparing musically trained and untrained individuals, while a few use a randomized experimental design, often randomly assigning musically untrained individuals to experiment and control groups (see S1 Table for an overview). Cross-sectional studies have yielded mixed findings on this relationship. Some studies demonstrate that adults with at least 6 years of instrumental music training [38–41, 45–48] and children with 3 years of extracurricular instruction [49] exhibit greater accuracy in identifying emotional prosodies. These benefits, however, do not extend to measures such as reaction times or emotional intensity ratings [39, 40, 50]. Contrarily, other research finds no significant benefit of music training in this domain, even with 5–14 years of instrumental practice [50–53]. These divergences may be attributable to varying sample sizes, impacting statistical significance, as well as the quantity of music training [54]. Specifically, studies which observed an effect of music training had larger sample sizes ($20 \leq n \leq 58$), making it easier to detect group differences. Meanwhile, studies which did not observe an effect of music training [50, 51, 53] had smaller sample sizes ($10 \leq n \leq 13$) and thus failed to obtain statistically significant results. This is because sample size is related to statistical significance ($p$-value); the smaller the sample size, the greater the $p$-value [55, 56]. However, the discrepancy between Mualem and Lavido's [52] study and other studies [45, 46] cannot be explained by sample size, and may be due to differences in the amount of music training. It is believed that at least 10 years of mixed music training is necessary to have an effect on auditory emotion recognition [54]. The average years of music training of the participants in the study of Mualem and Lavido was approximately 9 years, which may explain

why their performance was not altered. Notably, the above research predominantly concentrates on nontonal languages, leaving a significant gap in understanding the implications for tonal languages like Mandarin, Thai, or Vietnamese.

Recognizing that lexical tones are a common element in both language and music [57], existing studies highlight how language and cultural backgrounds influence the processing of emotional cues. Notable differences exist in how listeners of tonal and nontonal languages employ acoustic cues to interpret these tones [58, 59]. Furthermore, East Asians are more sensitive to verbal cues of an emotional utterance, whereas Westerners rely more on semantic cues for understanding and interpreting emotional information [60–62]. While longitudinal experimental studies have demonstrated that musical training enhances the recognition of emotional prosody in children [46, 63] and young adults [52, 64] but not in older adults [65, 66] who speak nontonal languages, the link between musical training and the perception of emotional prosody in speakers of tonal languages remains insufficiently established. To fill this gap in current research, the present study aims to explore the connection between music training and the recognition of emotional prosody in tonal languages. For this purpose, undergraduate students majoring in music performance or musicology, all of whom had received instrumental training since childhood, were compared with students without professional instrumental training. The focus was on their ability to recognize emotional prosody—specifically happiness, sadness, fear, anger, and neutrality—in Mandarin pseudosentences. Building on insights from existing studies [38–41], we hypothesized that music majors would excel in accurately identifying Mandarin emotional prosody compared to nonmusic majors.

## Method

### Participants

This study was approved by the Ethics Committee of the Harbin Conservatory of Music on 04/06/2023 and all participants provided verbal informed consent form and received monetary compensation for their participation. To achieve 90% power to detect a large main effect ($d = 0.8$) of group (music vs. nonmusic) at the 0.05 level, a minimum sample size of 34 for each group was required using G*Power (Version 3.1.9.7) [67]. Consequently, 36 music students and 36 nonmusic students who were native Mandarin speakers were recruited for the study from 04/08/2023 to 05/29/2023. The music students had been playing their chosen instruments for a minimum of 12 years, with an average of 13.9 hr of practice per week, or at least 3 hr of practice per day [54, 68]. The instruments covered a wide range, including piano ($n = 22$), violin ($n = 4$), double bass ($n = 1$), viola ($n = 1$), flute ($n = 1$), clarinet ($n = 1$), oboe ($n = 1$), harp ($n = 1$), yangqin ($n = 1$), trumpet ($n = 1$), suona ($n = 1$), and erhu ($n = 1$). In contrast, the nonmusic students had only received compulsory music education in school. To ensure differences in self-reported musical skills and behaviors between music and nonmusic students, the Goldsmiths Musical Sophistication Index (Gold-MSI) [69] was administered. As shown in Table 1, there is moderate evidence that the two groups were matched for age, sex (Fisher's exact test), years of education, nonverbal IQ as assessed by the Matrix Reasoning Item Bank (MaRs-IB) [70], and handedness as assessed by the Edinburgh Handedness Inventory [71]. However, there is very strong evidence that the music group had more years of training than the nonmusic group and scored higher on all Gold-MSI subtests as well as on the general musical sophistication factor. All participants had normal hearing and no history of neurological or psychiatric disorders.

### Stimuli

This study utilized emotional prosody materials drawn from the Chinese vocal emotional stimuli database [72]. The database features 874 pseudosentences, produced by four native

**Table 1. Participant characteristics.**

| Variable | Music group | Nonmusic group | Statistic | Effect size | 95% CI | BF |
|---|---|---|---|---|---|---|
| Age (years) | 22.08 ± 2.17 | 22.31 ± 1.95 | $t(69.25) = -0.46$ | $g^* = -0.11$ | [−0.56, 0.35] | 0.13 |
| Sex (female/male) | 34/2 | 34/2 | – | $\log OR = 0$ | [−2.02, 2.02] | 0.14 |
| Education (years) | 15.94 ± 2.34 | 16.11 ± 2.04 | $t(68.67) = -0.32$ | $g^* = -0.08$ | [−0.53, 0.38] | 0.12 |
| MaRs-IB (%) | 77.53 ± 10.95 | 77.42 ± 11.45 | $t(69.86) = 0.04$ | $g^* = 0.01$ | [−0.45, 0.47] | 0.12 |
| Handedness | 81.02 ± 16.89 | 83.27 ± 15.19 | $t(69.22) = -0.59$ | $g^* = -0.14$ | [−0.60, 0.32] | 0.14 |
| Years of music training | 15.42 ± 2.25 | – | $t(35) = 41.16$ | $g = 6.71$ | [5.11, 8.27] | $6.27 \times 10^{27}$ |
| **Gold-MSI** | | | | | | |
| Musical Training | 5.56 ± 0.47 | 1.31 ± 0.25 | $t(53.69) = 48.22$ | $g^* = 11.21$ | [8.99, 13.32] | $\infty$ |
| Perceptual Abilities | 5.98 ± 0.67 | 3.73 ± 0.95 | $t(62.95) = 11.65$ | $g^* = 2.71$ | [2.05, 3.37] | $3.61 \times 10^{28}$ |
| Singing Abilities | 5.33 ± 0.74 | 2.67 ± 1.07 | $t(62.36) = 12.29$ | $g^* = 2.86$ | [2.18, 3.53] | $7.50 \times 10^{31}$ |
| Active Engagement | 5.24 ± 0.68 | 3.10 ± 1.02 | $t(60.88) = 10.54$ | $g^* = 2.45$ | [1.82, 3.08] | $1.55 \times 10^{23}$ |
| Emotions | 5.76 ± 0.73 | 4.35 ± 0.97 | $t(64.84) = 6.97$ | $g^* = 1.62$ | [1.08, 2.15] | $4.07 \times 10^{9}$ |
| General Sophistication | 5.40 ± 0.51 | 2.58 ± 0.77 | $t(60.60) = 18.38$ | $g^* = 4.28$ | [3.39, 5.16] | $2.65 \times 10^{72}$ |

Mandarin-Chinese speakers (two male), each conveying one of seven emotions (happiness, sadness, fear, anger, disgust, pleasant surprise, and neutrality) through prosodic variation in 35 unique utterances. Each utterance, ranging from 1 to 2 s in duration, adhered to Chinese grammatical rules while remaining semantically meaningless to eliminate the influence of semantics on emotional prosody perception [72–74]. The stimuli were validated with a group of native Mandarin speakers who participated in a seven-option forced choice task to identify the expressed emotion and a 5-point Likert item (1 = *very weak*, 5 = *very intense*) to rate its intensity, excluding neutrality. Stimuli meeting satisfactory identification accuracy criteria were included in the database; specifically, those exceeding a 42.86% average recognition rate (3 times the chance level) for the target emotion and remaining below 42.86% for any nontarget emotion [72, 75, 76].

Twenty-one utterances each expressing happiness, sadness, fear, anger and neutrality were selected from recordings of a male and a female speaker, yielded 42 stimuli per emotion. Table 2 presents the recognition rate and intensity rating for each emotion except neutrality. A one-way analysis of variance (ANOVA) was conducted on the average accuracy rates. Although normality assumptions were met ($Ws \geq 0.94$, $ps \geq .200$), sphericity was violated

**Table 2. Acoustic and perceptual measures for each emotion category averaged across two speakers ($n = 21$).**

| Measure | Happiness | | Sadness | | Fear | | Anger | | Neutrality | |
|---|---|---|---|---|---|---|---|---|---|---|
| | M | SD | M | SD | M | SD | M | SD | M | SD |
| **Acoustic features** | | | | | | | | | | |
| $f_0$ mean (Hz) | 1.58 | 0.13 | 0.78 | 0.07 | 1.14 | 0.13 | 1.54 | 0.15 | 0.40 | 0.06 |
| $f_0$ range (Hz) | 2.25 | 0.32 | 0.84 | 0.31 | 1.22 | 0.26 | 2.05 | 0.32 | 0.93 | 0.24 |
| Amp mean (dB) | 0.57 | 0.04 | 0.63 | 0.05 | 0.58 | 0.05 | 0.55 | 0.04 | 0.56 | 0.05 |
| HNR mean (dB) | 12.16 | 1.43 | 14.55 | 1.23 | 11.92 | 1.77 | 9.22 | 1.56 | 10.15 | 1.48 |
| HNR SD (dB) | 5.81 | 0.65 | 6.84 | 0.54 | 6.87 | 0.73 | 5.55 | 0.56 | 6.52 | 0.71 |
| Speech rate (syllables/s) | 5.99 | 0.65 | 5.12 | 0.40 | 5.88 | 0.41 | 7.01 | 0.66 | 6.23 | 0.47 |
| **Perceptual features** | | | | | | | | | | |
| Recognition rate | 0.72 | 0.10 | 0.83 | 0.05 | 0.77 | 0.11 | 0.81 | 0.07 | 0.83 | 0.09 |
| Intensity rating | 2.64 | 0.33 | 3.15 | 0.26 | 2.94 | 0.28 | 3.01 | 0.19 | – | – |

The $f_0$ and Amp values in the original data set were normalized.

($W$ = 0.26, $p$ = .003). Therefore, the degrees of freedom were corrected using the Greenhouse-Geisser method ($\hat{\varepsilon}$ = 0.709). The analysis revealed moderate evidence for a main effect of emotion with a large effect size, $F(2.83, 56.70)$ = 5.87, $\omega_p^2$ = 0.185, 95% CI [0.018, 0.343], $BF$ = 6.09. Pairwise comparisons (see S2 Table) showed that happiness was recognized less accurately than sadness, anger, and neutrality. No differences were found between any other emotion pairs. Similar analyses were conducted for the average intensity ratings. The data met normality assumptions ($Ws \geq 0.94$, $ps \geq .223$) but violated sphericity ($W$ = 0.55, $p$ = .048). ANOVA with the Greenhouse-Geisser correction ($\hat{\varepsilon}$ = 0.703) revealed very strong evidence for a main effect of emotion with a large effect size, $F(2.11, 42.20)$ = 14.50, $\omega_p^2$ = 0.386, 95% CI [0.150, 0.557], $BF$ = $1.52 \times 10^3$. Pairwise comparisons (see S2 Table) indicated that happiness was rated as less intense than sadness, fear, and anger, while sadness was rated as more intense than fear. No differences were found between anger and sadness or fear.

To assess whether the main acoustic features (mean fundamental frequency [$f_0$ mean], $f_0$ range, mean amplitude [Amp mean], mean harmonics-to-noise ratio [HNR mean], HNR variation [HNR $SD$], and speech rate) of the stimuli (see Table 2) could predict the intended emotion categories, we conducted a discriminant analysis following the procedures of Liu and Pell [72] and Paulmann and Uskul [77]. Using IBM SPSS Statistics (Version 29.0.2), we treated the main acoustic features as predictor variables and the intended emotions as the outcome variable for the pseudosentences. The analysis revealed that 84.76% (178/210) of the pseudosentences were correctly classified in terms of their intended emotion: happiness 78.57% (33/42), sadness 97.62% (41/42), fear 76.19% (32/42), anger 80.95% (34/42), and neutrality 90.48% (38/42). These results suggest that listeners can reliably identify the intended emotions based on the analyzed acoustic features.

## Measures

**MaRs-IB.** The MaRs-IB was employed to measure participants' nonverbal IQ [70]. This open-access online assessment has demonstrated good reliability and validity and is similar to Raven's Advanced Progressive Matrices [78]. The test consists of 80 items, each featuring a 3×3 matrix in which eight out of nine cells contain abstract shapes that vary along four dimensions: color, size, shape and position. Participants must select the missing shape from an array of four options. Each item answered correctly is worth one point, while incorrect items are worth zero points. A higher score or percentage of accuracy is indicative of a stronger spatial reasoning ability and a higher nonverbal IQ [79].

**Gold-MSI.** The Gold-MSI [69] is a questionnaire designed to assess individual differences in self-reported musical skills and behaviors among both musicians and the general population. It consists of 38 items divided into five subscales: Musical Training (7 items), Perceptual Abilities (9 items), Singing Abilities (7 items), Active Engagement (9 items), and Emotions (6 items). Respondents are asked to rate their agreement with each statement on a 7-point Likert scale (from 1 *strongly disagree* to 7 *strongly agree*) for the first 31 items or select one of seven options for the remaining 7 items. The scores for each dimension are then summed, with higher scores indicating a higher level of musical skills and greater musical engagement or emotional experiences. Additionally, the General Sophistication consists of 18 items, with higher scores indicating a greater degree of musical sophistication. The Chinese version of the Gold-MSI has been found to demonstrate high reliability and validity [80, 81] and was used in the present study. In the current study, the reliability of the Musical Training, Perceptual Abilities, Singing Abilities, Active Engagement, Emotions, and General Sophistication factors, as evaluated by McDonald's ω, was 0.98 (95% CI [0.97, 0.99]), 0.95 (95% CI [0.93, 0.96]), 0.94 (95% CI [0.91, 0.96]), 0.93 (95% CI [0.91, 0.96]), 0.84 (95% CI [0.78, 0.90]), and 0.97 (95% CI [0.96, 0.98]) respectively.

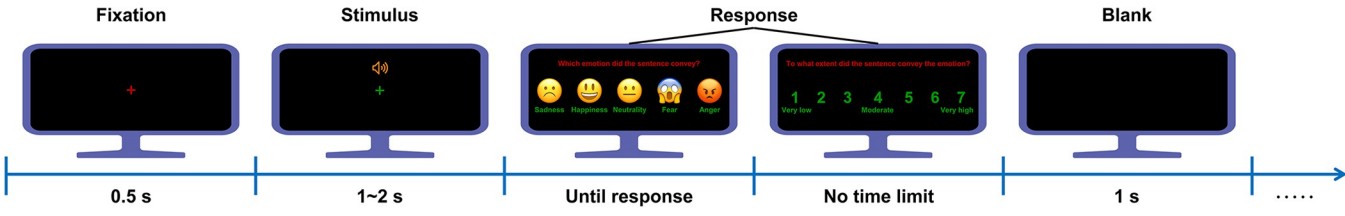

**Fig 1. Timing of one trial used in the experiment.**

## Procedure

Participants completed the online experiment through the NAODAO platform (www.naodao. com) between 04/15/2023 and 06/06/2023. Before the experiment began, participants were required to read and fully understand the informed consent form. The experiment only commenced automatically upon their explicit confirmation of participation. To ensure participants were familiar with the procedure, a practice session preceded the formal experiment. This practice session included four practice stimuli and feedback on accuracy. Additionally, participants could adjust the headphone volume to a comfortable level.

Fig 1 displays the design of the trial in the experiment. At the start of each trial, a red fixation cross appeared in the center of the screen for 0.5 s to indicate that a stimulus was about to come. The fixation cross then turned green as the stimulus began to play for approximately 1–2 s. After listening to the stimulus, a response interface appeared, and participants were asked to click on one of the five emotional prosody labels and icons to match the emotion expressed in the speech as quickly and accurately as possible. Subsequently, a 7-point Likert item of emotion intensity (1 = *very low*, 7 = *very high*) was presented on the screen, and participants were asked to rate the intensity of the emotion in the stimulus without a time limit. To control for potential sex-related differences in vocal acoustic features [82, 83] and reduce participants' cognitive load from sudden change in speaker sex, we presented sentences spoken by male and female speakers in two separate blocks, with 105 randomly presented sentences per block. The order of the blocks was counterbalanced across participants, with a 1–2 min break between the two blocks.

## Data analysis

Before data analysis, trials with response times (RTs) falling below 0.2 s (anticipatory responses or fast guesses) or exceeding 2 *SD* above the mean for each participant (distraction) were excluded as outliers. Trials with errors were also omitted for RT and intensity ratings. However, in speeded decision tasks, a pervasive speed-accuracy tradeoff (SAT) [84, 85] exists. Analyzing mean RT or accuracy rates separately could, therefore, lead to contradictory or invalid conclusions [86–88]. To control for this potential SAT, a composite metric incorporating speed and accuracy should be used for performance evaluation. Composite metrics offer more sensitive and reliable assessments than relying solely on accuracy or RT measures [89, 90].

Several such metrics exist, including inverse efficiency score, rate correct score, linear integrated speed-accuracy score, and balanced integration score (BIS). Among these, only the BIS exhibits relatively insensitivity to SAT [91, 92]. To minimize the SAT effect and preserve "true" effects, we opted for the BIS, assigning equal weights to speed and accuracy. The BIS is calculated by subtracting the standardized (i.e., *z* scores) mean correct RT from the standardized mean proportions of correct responses across all conditions and participants [91–93]. Previous studies have widely adopted this metric [e.g., 94–102]. Using this approach, participants achieving high accuracy at the expense of RT or fast RTs at the expense of accuracy will have

BISs near zero compared to the sample average. Participants with both high accuracy and fast RTs will have higher positive BISs, while those with low accuracy and slow RTs will have lower negative BISs. Therefore, higher BIS represents faster and more accurate performance. To investigate the SAT in this study, we first analyzed accuracy and RT data. If an SAT is identified, we will subsequently analyze the data using the BIS framework. Both the frequentist and Bayesian approaches were applied to the data.

**Frequentist analysis.** In frequentist analysis, effect sizes and confidence intervals are used to evaluate the magnitude and precision of estimation of an effect and to draw conclusions, instead of relying on *p*-values from traditional null hypothesis significance tests [103–108]. This method of interpretation is endorsed by the American Psychological Association [109, 110] and several journals [111–113] and researchers [114–117]. If the confidence interval for an effect size does not include zero, the effect is considered to be present; while if the confidence interval (CI) does include zero, the effect is considered to be nonexistent [118–121]. Additionally, effect sizes can be categorized as small, medium, or large based on established thresholds: *g* or *d* = 0.20, 0.50, and 0.80 [122], *r* = 0.10, 0.30, and 0.50 [123], and $\omega^2$ = 0.01, 0.06, and 0.14 [124]. This study employed effect sizes and their confidence intervals to interpret the results.

We analyzed the demographic variables of age, years of education, MaRs-IB, handedness, and Gold-MSI using Welch's *t*-tests with JASP (Version 0.19.0). Hedges' *g** and their 95% CIs [125] were calculated using the *deffectsize* package (Version 0.0.0.9000) in R (Version 4.4.1) and RStudio (Version 2024.04.2+764). For sex, we used Fisher's exact test in JASP and calculated log *OR* and its 95% CI. For years of music training, we performed a one-sample *t*-test using the *statsExpressions* package (Version 1.5.5) and calculated *g* and its 95% CI.

For each perceptual feature, we conducted an ANOVA with emotion (happiness, sadness, fear, anger, and/or neutrality) as a within-items variable using jamovi (Version 2.5.6). Prior to the ANOVAs, we tested for normality with the Shapiro-Wilk test and sphericity with Mauchly's test. If assumptions were met, parametric tests were conducted. If any assumption was violated, the Greenhouse-Geisser correction ($\hat{\varepsilon}$) was applied to adjust the degrees of freedom.

BIS was used to assess overall task performance, but we also conducted a correlation analysis between accuracy and RT to test the SAT. We used repeated measures correlation [126–128] implemented in the *rmcorr* package (Version 0.7.0). The effect size (correlation coefficient) and its 95% CI were estimated using the bootstrapping with the suggested number of 10000 bootstrap resamples [129–132].

For accuracy, RT, BIS and emotional intensity, we conducted a two-way mixed design ANOVA with group (music and nonmusic) as the between-subjects variable and emotion (happiness, sadness, fear, anger, and neutrality) as the within-subjects variable. Data were tested for normality, homogeneity of variance with Levene's test, and sphericity using JASP. Parametric tests were conducted if assumptions were met; otherwise, nonparametric tests were conducted with *ARTool* (Version 0.11.1). Post hoc comparisons with Bonferroni correction were performed using the ART-C program for significant interaction effects. We used the *effectsize* package (Version 0.8.9) to calculate $\omega_p^2$ and its 95% CI for each ANOVA effect, and *d* and its 95% CI for each pairwise or post hoc comparison.

**Bayesian analysis.** To assess the strength of the evidence and obtain more conservative results, we conducted Bayesian hypothesis testing. The Bayes factor (*BF*) was calculated to quantify the relative plausibility of the null hypothesis ($H_0$) and alternative hypothesis ($H_1$) given the observed data [133–136]. Conventionally, a *BF* of 1–3 indicates weak evidence for $H_1$, 3–10 indicates moderate evidence, 10–30 indicates strong evidence, and above 30 indicates very strong evidence for $H_1$. Conversely, *BF*s between 0.33–1, 0.10–0.33, 0.03–0.10, and below

0.03 indicate weak, moderate, strong, and very strong evidence for $H_0$, respectively [137, 138]. While acknowledging the inherent subjectivity in interpreting evidential strength, this study adopted the common practice of considering moderate evidence as a sufficient threshold for drawing meaningful conclusions. For example, a *BF* of 5 indicates that the data are 5 times more likely under $H_1$ than $H_0$, implying substantial support for the presence of an effect.

For age, years of education, MaRs-IB, handedness and Gold-MSI, we employed the Bain module in JASP to perform Bayesian Welch's *t*-tests [139] and calculate *BF*s. For sex, we employed JASP to perform a Bayesian contingency table test based on an independent multi-nomial sampling scheme [140] and obtain the *BF*. For years of music training, we used JASP to execute a Bayesian one-sample *t*-test with the default prior (Cauchy $r = 0.707$) and determine the *BF*.

For perceptual features, accuracy, RT, BIS, or emotional intensity, we calculated *BF*s using *anovaBFcalc* (Version 0.1.0) with default $\alpha = -0.5$ (MWS method) based on *F*-values and degrees of freedom obtained from frequentist analysis. For pairwise or post hoc comparisons, we computed *BF*s using the *BayesFactor* package (Version 0.9.12–4.7) with a prior scale factor of $r = 0.707$, based on *t* values and sample sizes obtained from frequentist analysis.

## Results

We removed 24 music group trials (0.16%) and 86 nonmusic group trials (0.57%) with RT < 0.2 s. Subsequently, we discarded 312 music group trials (2.08%) and 368 nonmusic group trials (2.45%) with RT > (*M* + 2 *SD*) s. Finally, we eliminated 753 music group trials (5.25%) and 1742 nonmusic group trials (12.16%) due to error responses.

### Accuracy and response time

Regarding the accuracy data, all dataset except for happiness in the nonmusic group ($W = 0.95$, $p = .146$) deviated from normality ($Ws \leq 0.93$, $ps \leq .025$). Additionally, all datasets except for happiness ($F(1, 70) = 2.40$, $p = .126$) violated the assumptions of homogeneity of variance ($Fs(1, 70) \geq 5.01$, $ps \leq .028$) and sphericity ($W = 0.54$, $p < .001$). As for the RT data, all datasets except for happiness ($W = 0.95$, $p = .138$) and sadness ($W = 0.95$, $p = .121$) in the nonmusic group violated the normality assumption ($Ws \leq 0.93$, $ps \leq .024$). Furthermore, all RT datasets violated the assumptions of homogeneity of variance ($Fs(1, 70) \geq 4.47$, $ps \leq .038$) and sphericity ($W = 0.65$, $p < .001$).

Fig 2 illustrates the accuracy and RTs of the music and nonmusic groups across all emotions. It is evident that the music group exhibited higher accuracy but slower RTs compared to the nonmusic group for all emotions. To validate these visual observations, separate two-way ANOVAs were conducted on the accuracy and RT data. There was very strong evidence for a large main effect of group on accuracy ($F(1,70) = 62.10$, $\omega_p^2 = 0.459$, 95% CI = [0.290, 0.589], $BF = 3.13 \times 10^8$), indicating that the music group ($M = 89.47$, $SD = 11.67$) performed better than the nonmusic group ($M = 75.39$, $SD = 20.34$). Similarly, there was very strong evidence for a large main effect of group on RTs ($F(1,70) = 14.33$, $\omega_p^2 = 0.156$, 95% CI = [0.033, 0.312], $BF = 59.08$), with the music group exhibiting slower RTs ($M = 1.28$, $SD = 0.60$) than the nonmusic group ($M = 0.89$, $SD = 0.37$). These results suggest the presence of an SAT for both music and nonmusic subjects.

Fig 3 depicts the speed–accuracy correlations for the combined group and each individual group across all emotions. A moderate negative correlation between RT and accuracy was observed for the combined group, $r(287) = -0.397$, 95% CI [−0.485, −0.306], consistent with the SAT phenomenon. This correlation remained similar for the music group, $r(143) = -0.366$, 95% CI [−0.504, −0.204], but was larger for the nonmusic group, $r(143) = -0.510$, 95% CI

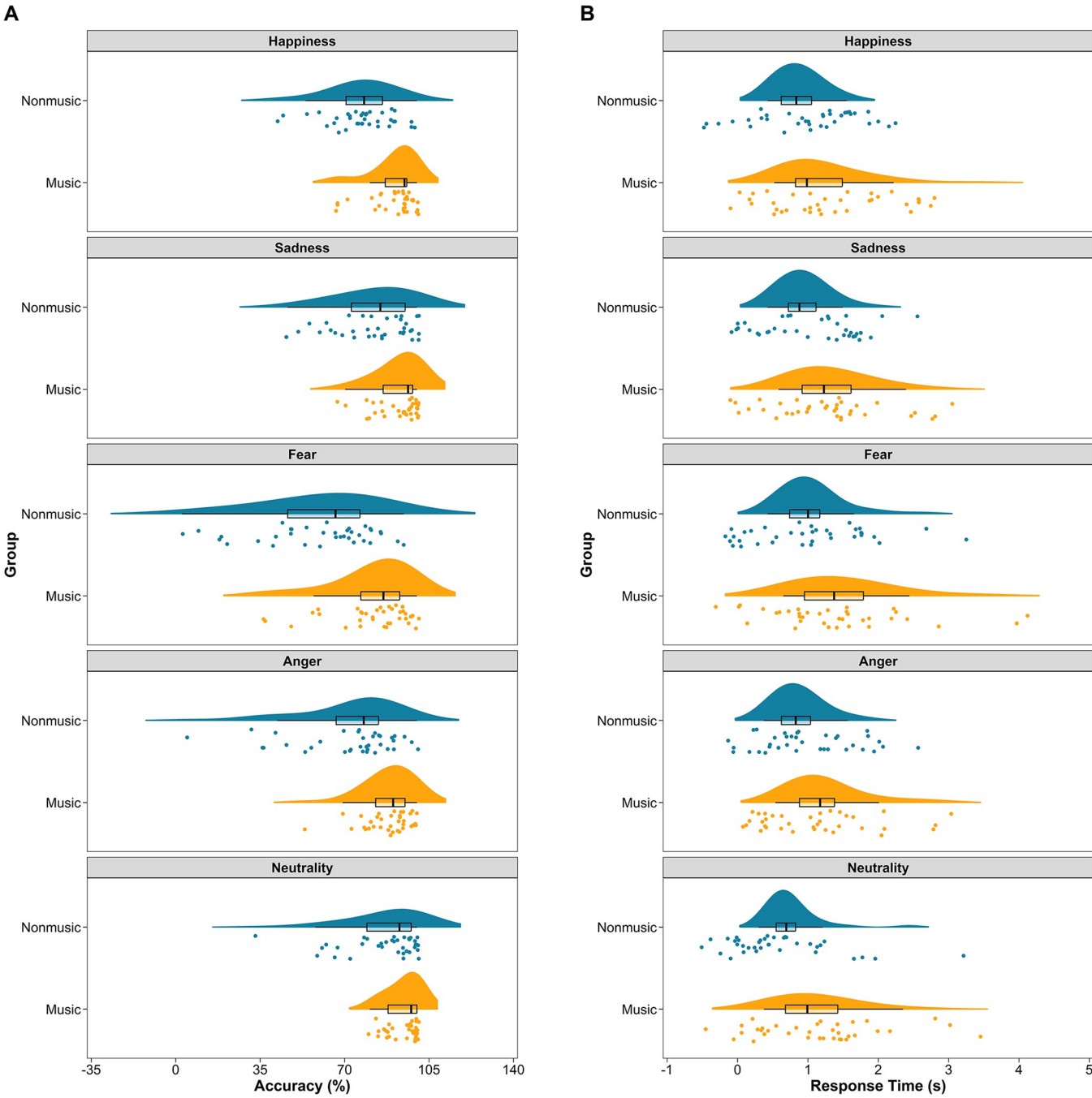

**Fig 2.** Accuracy (A) and response time (B) as a function of group and emotion. These raincloud plots, combining a halved violin plot, a boxplot, and jittered individual data points, depict the data distribution. They were generated using *ggplot2* (Version 3.5.1), *PupillometryR* (Version 0.0.5), *ggpp* (Version 0.5.8–1), and *cowplot* (Version 1.1.3) in R.

[−0.616, −0.392]. These findings suggest that faster RT generally come at the expense of reduced accuracy within individuals, highlighting the suitability of BIS as a more comprehensive measure of task performance.

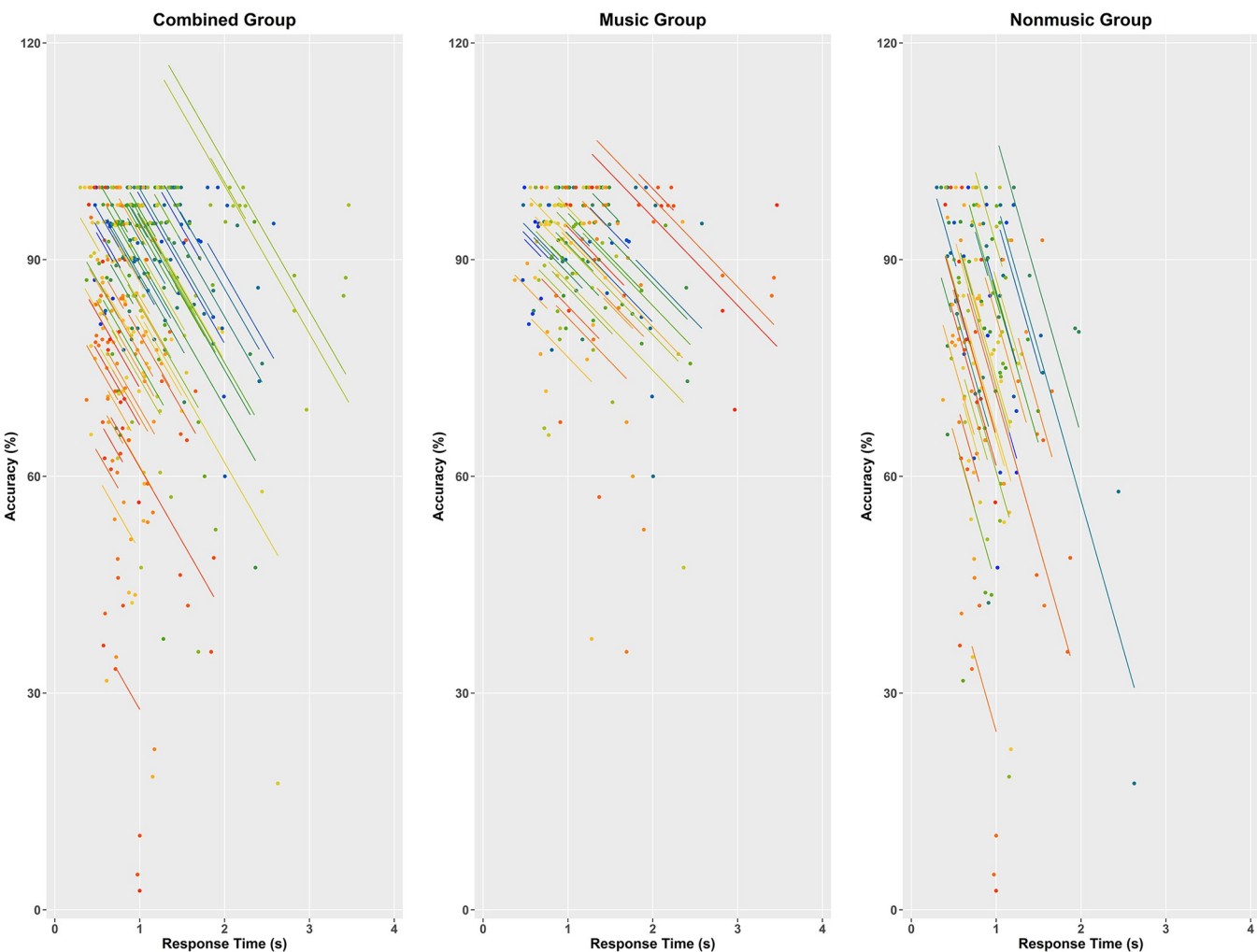

**Fig 3. Scatterplots depicting correlations between response time and accuracy for the combined group and each group across emotion.** Each dot represents the average response time and accuracy for a specific emotion. Participant identity is indicated by color, and colored lines show the repeated measures correlation fits for each participant's data. These scatterplots were created using the following R packages: *rmcorr* (Version 0.7.0), *ggplot2* (Version 3.5.1), *pals* (Version 1.9), and *cowplot* (Version 1.1.3).

## Balanced integration score

All but one dataset (happiness for the nonmusic group: $W = 0.96$, $p = .281$) deviated from normality ($Ws \leq 0.94$, $ps \leq .037$). Homogeneity of variance was confirmed for all datasets ($Fs(1, 70) \leq 0.76$, $ps \geq .388$), but the data violated the assumption of sphericity ($W = 0.62$, $p < .001$).

Fig 4A illustrates the BIS of the music and nonmusic groups across all emotions, suggesting comparable performance. A nonparametric ANOVA corroborated these observations. There was substantial evidence for the absence of a main effect of group, $F(1,70) = 0.14$, $\omega_p^2 = 0.000$, 95% CI = [0.000, 0.000], $BF = 0.103$, indicating that the music group ($M = 0.03$, $SD = 1.40$) did not outperform the nonmusic group ($M = -0.03$, $SD = 1.19$). In contrast, there was very strong evidence for a large main effect of emotion, $F(4, 280) = 34.62$, $\omega_p^2 = 0.321$, 95% CI = [0.230, 0.396], $BF = 8.41 \times 10^{19}$. Pairwise comparisons (see S3 Table) showed that participants performed best in detecting neutral emotional prosody ($M = 0.74$, $SD = 1.19$), followed by sadness ($M = 0.19$, $SD = 1.20$), happiness ($M = 0.17$, $SD = 1.08$), or anger ($M = -0.08$, $SD = 1.40$), and worst in detecting fear ($M = -1.03$, $SD = 1.70$). Notably, there was very strong evidence for the

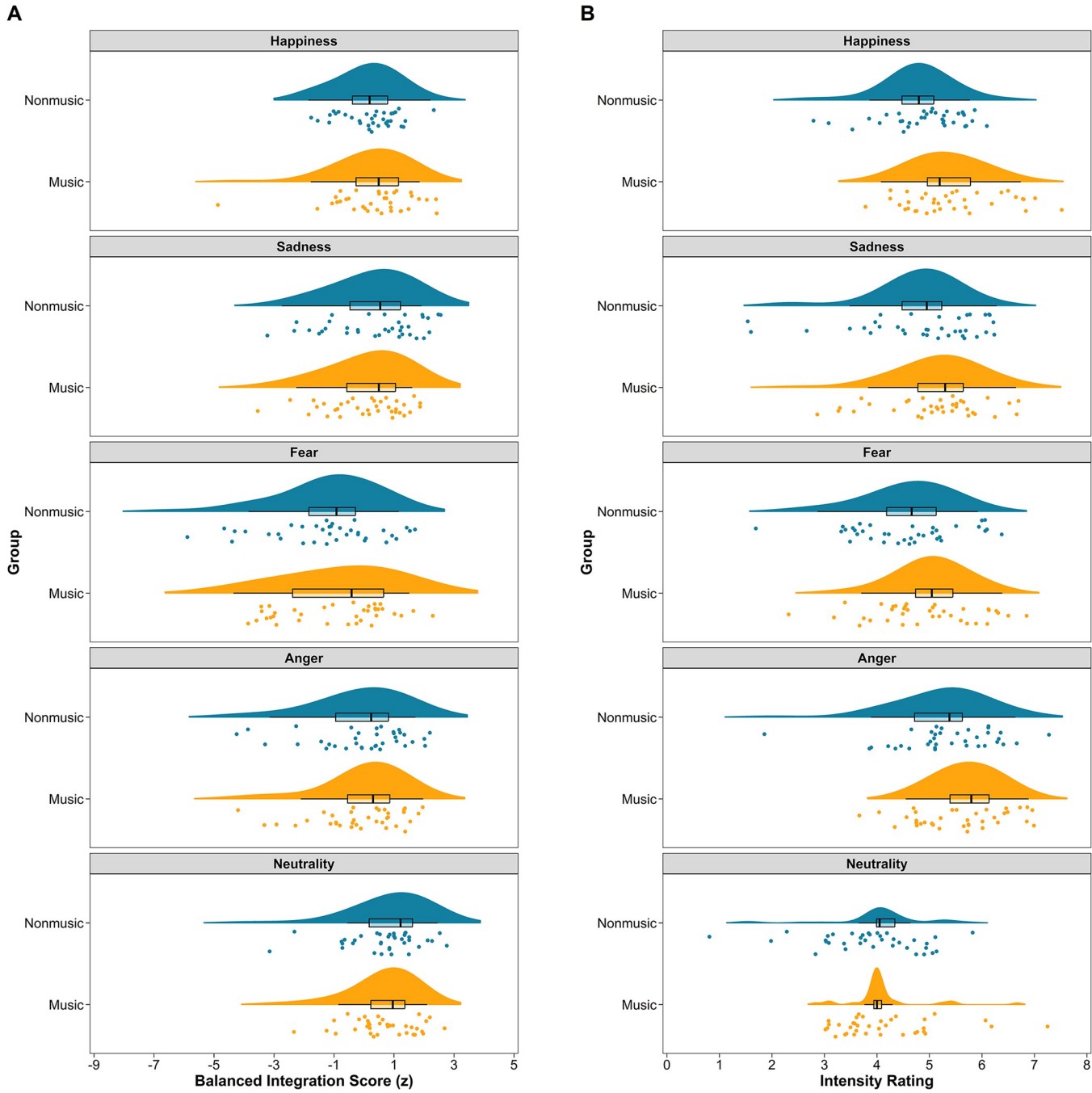

**Fig 4.** Balanced integration score (A) and intensity ratings (B) as a function of group and emotion. These raincloud plots, combining a halved violin plot, a boxplot, and jittered individual data points, depict the data distribution. They were generated using *ggplot2* (Version 3.5.1), *PupillometryR* (Version 0.0.5), *ggpp* (Version 0.5.8–1), and *cowplot* (Version 1.1.3) in R.

absence of an interaction effect between group and emotion, $F(4, 280) = 2.07$, $\omega_p^2 = 0.015$, 95% CI = [0.000, 0.041], $BF = 2.21 \times 10^{-3}$. These findings suggest that instrumental training is not associated with the ability to recognize emotional prosody in tonal languages.

**Table 3. Post hoc comparisons for the Group × Emotion interaction on intensity ratings.**

| Emotion | Music group | Nonmusic group | $t(189.57)$ | $d$ | 95% CI | BF |
|---------|-------------|----------------|-------------|-----|--------|-----|
| Happiness | 5.32 ± 0.62 | 4.77 ± 0.69 | 3.51 | 0.51 | [0.22, 0.80] | 37.43 |
| Sadness | 5.16 ± 0.82 | 4.77 ± 0.82 | 2.37 | 0.34 | [0.06, 0.63] | 2.58 |
| Fear | 5.03 ± 0.67 | 4.59 ± 0.78 | 2.57 | 0.37 | [0.09, 0.66] | 3.87 |
| Anger | 5.72 ± 0.56 | 5.21 ± 0.90 | 2.95 | 0.43 | [0.14, 0.72] | 9.03 |
| Neutrality | 4.11 ± 0.67 | 4.06 ± 0.86 | −0.58 | −0.08 | [−0.37, 0.20] | 0.28 |

Only the $M \pm SD$ was computed from the raw data; other statistics were derived from the aligned-and-ranked data.

### Intensity ratings

All but four datasets (neutrality for the music group, and sadness, anger, and neutrality for the nonmusic group: $Ws \leq 0.90$, $ps \leq .004$) exhibited normality ($Ws \geq 0.94$, $ps \geq .060$). Homogeneity of variance was met for all datasets ($Fs(1, 70) \leq 3.02$, $ps \geq .087$), but the assumption of sphericity was violated ($W = 0.35$, $p < .001$).

Fig 4B demonstrates the intensity ratings of the music and nonmusic groups across all emotions, suggesting higher ratings for the music group. A nonparametric ANOVA revealed moderate evidence for a medium main effect of group, $F(1,70) = 8.47$, $\omega_p^2 = 0.094$, 95% CI = [0.006, 0.240], $BF = 4.92$. This indicates that the music group ($M = 5.07$, $SD = 0.85$) assigned higher intensity ratings than the nonmusic group ($M = 4.68$, $SD = 0.89$). Additionally, a very strong evidence for a large main effect of emotion emerged, $F(4, 280) = 74.55$, $\omega_p^2 = 0.508$, 95% CI = [0.428, 0.571], $BF = 3.24 \times 10^{39}$. Pairwise comparisons (see S3 Table) showed that angry emotional prosody received the highest rating ($M = 5.47$, $SD = 0.79$), followed by happy ($M = 5.05$, $SD = 0.71$) and sad ($M = 4.97$, $SD = 0.84$) or fearful ($M = 4.81$, $SD = 0.75$), with neutral receiving the lowest rating ($M = 4.09$, $SD = 0.77$). Furthermore, moderate evidence was found for a medium interaction effect between group and emotion, $F(4, 280) = 6.15$, $\omega_p^2 = 0.067$, 95% CI = [0.014, 0.121], $BF = 4.80$. Post hoc comparisons (Table 3) revealed very strong evidence for higher ratings by the music group for happy emotion, moderate evidence for higher ratings for fearful and ***angry*** emotions, similar ratings for neutral emotion, and only weak evidence for higher ratings for sad emotion. These findings suggest that instrumental training is associated with heightened sensitivity to the intensity of emotional prosodies in tonal languages.

### Discussion

This study uniquely explores the link between instrumental music training and emotional prosody identification in tonal languages, focusing on Mandarin. Utilizing robust analysis with effect size estimates and Bayesian factors, we draw strong conclusions based on our sizable sample of 72 participants. Interestingly, the results revealed no difference in emotional prosody identification between musically trained and untrained participants. However, music majors assigned higher intensity ratings to happy, fearful, and angry prosodies. These findings suggest that music training is not positively correlated with an improved ability to recognize emotional prosody in tonal languages but is instead associated with perceived emotional intensity. This adds a new dimension to our understanding of an intricate relationship between music training and emotion perception in tonal languages, distinct from research focused on nontonal languages.

Utilizing the BIS to mitigate speed-accuracy tradeoff effects in our data, we discovered that music and nonmusic majors exhibited similar proficiency in recognizing emotional prosodies.

This outcome contrasts with previous studies suggesting that instrumentalists [38–41, 48] and mixed musicians [45, 46, 49] exhibit superior performance on emotional prosody recognition in nontonal languages. This inconsistency may be explained by the intricate interplay between music and language experience in shaping emotional prosody recognition. Indeed, prior research has demonstrated that musicians with a nontonal language background (i.e., English) outperform their nonmusician counterparts in tasks involving linguistic pitch perception [141, 142] and lexical tone perception [14, 143–145]. However, this advantage is not as pronounced among musicians with a tonal language background such as Mandarin [146], Cantonese [147], and Thai [144, 148]. The null result observed in this study may be related to the characteristics of the music subjects. Specifically, the music subjects in our study are more likely to possess absolute pitch—the ability to name a tone without reference—due to their experience with a tonal language. Although we did not directly assess absolute pitch ability, previous research indicates that 53%-72% of Chinese music students possess absolute pitch [149–152]. Given that emotional prosody perception primarily relies on relative pitch processing [153, 154], which focuses on the pitch changes and patterns rather than absolute pitch, the presence of absolute pitch ability may not necessarily confer an advantage in emotional prosody recognition tasks. In contrast, among musical subjects who spoke a nontonal language, only 3%-10% have absolute pitch [149, 151, 155, 156], while the majority possess relative pitch. This heightened capacity for relative pitch processing [157, 158] likely contributes to their enhanced recognition of emotional prosody.

We also discovered that music majors gave higher intensity ratings to happy, fearful, and angry emotional prosody compared to nonmusic majors. This observation aligns with previous evidence that musicians provided higher intensity ratings for happy emotional prosody in English than did nonmusicians [159]. However, it is contradictory to the studies reporting no differences in intensity or arousal ratings for emotional prosody in Portuguese or German between musicians and nonmusicians [40, 160]. The discrepancy may be attributed to the difference in sample sizes; our study included 36 music and 36 nonmusic subjects, whereas Dibben et al.'s and Lima and Castro's studies involved 17 music and 15 nonmusic subjects, and 20 music and 20 nonmusic subjects, respectively. The larger sample size in our study, being at least 1.8 times larger than those of the other studies, likely facilitated a more sensitive detection the effect of music training. Our findings might be explained by enhanced auditory processing and selective attention abilities in music subjects. Through extensive instrument training and practice, music subjects develop advanced auditory skills that make them more sensitive to subtle changes in acoustic features (e.g., pitch, intensity, and tempo) shared by speech and music. Studies show musicians excel in pitch [161–165], intensity [166], and tempo [167, 168] discrimination compared to nonmusicians. High-arousal emotional speech (e.g., happiness, fear, and anger), often characterized by high pitch and/or large pitch variability, rapid speech rate, and/or increased intensity [18, 21], exhibits more pronounced shifts in these features. As a result, even slight increases in these features might be more salient to music subjects, potentially leading to higher intensity ratings. Furthermore, high-arousal emotional prosodies tend to capture more attention than low-arousal ones. Musicians typically demonstrate superior auditory attentional capacities relative to nonmusicians [169–171]. This heightened selective attention allows them to allocate more cognitive resources to processing emotional information in high-arousal speech prosody. This deeper processing amplifies the perceived intensity of the emotion. This dual effect of enhanced perception and attentional capacity may result in a heightened emotional experience, making high-arousal emotions seem more intense than those with low arousal.

Music majors in our study displayed similar levels of emotion recognition as nonmusic majors but provided higher intensity ratings for emotional prosodies in Mandarin. This

dissociation pattern may arise from differences in task demands between emotion recognition and intensity ratings. Emotion recognition tasks, such as categorizing emotions, are generally regarded as explicit tasks; intensity rating tasks are seen as more implicit [172–174]. This distinction stems from the fact that categorizing emotions require direct access to a stable internal representation of a stimulus and a comparison of the presented stimulus with mental prototypes of each emotion. This process necessitates higher activation levels and a conscious representation of the stimulus. On the other hand, rating the intensity of an emotion may not require verbal categorization of a given emotion and relies on a weaker internal representation of the stimulus, with less precise knowledge about this stimulus being necessary for a more global judgement [173–175]. The dissociation between emotion categorization and emotion intensity judgment has been documented in the processing of musical [173, 175–177] and vocal [174, 178] emotion. Thus, our findings suggest that instrumental music training may primarily relate to the implicit, rather than explicit, processing of emotional prosody in tonal languages. This cross-domain transfer likely stems from a statistical learning mechanism employed in music and language acquisition. Statistical learning refers to a domain-general perceptual mechanism where learners unconsciously extract statistical patterns or regularities from sensory input over time, without explicit instruction [29, 179]. This type of learning is particularly relevant for implicit knowledge acquisition that relies on specific acoustic cues [180]. Therefore, even passive exposure to music allows both music and nonmusic subjects to acquire implicit knowledge about the statistical regularities between acoustic cues and musical emotions [181, 182]. On the basis of this implicit knowledge, all subjects are likely to show heightened attention to these cues in emotional prosody and make intensity judgments. Nonetheless, music training enhances music subjects' sensitivity to the acoustic cues [183, 184], leading to a superior ability to attend to them compared to nonmusic subjects. This heightened sensitivity arises from the focused and intensive nature of music training, which makes music subjects particularly adept at detecting the underlying statistical regularities. Indeed, research demonstrates that musicians outperform nonmusicians in auditory statistical learning tasks involving pitch [161, 185–187] and rhythm [188–190]. Considering the implicit nature of statistical learning [191], its impact is more likely to be observed in implicit tasks rather than explicit ones.

This study has several limitations that warrant careful consideration. First, the quasi-experimental design precludes establishing a causal link between instrumental music training and emotional prosody processing. Future studies should employ randomized experimental designs to address this issue. Second, limited sample size likely contributed to the weakness of evidence for some findings. Consequently, subsequent future studies should use larger sample sizes to obtain more reliable results. Third, this investigation utilized only one tonal language, raising questions about the generalizability of the findings to other tonal languages. Future research is needed to explore this broader applicability. Finally, the mechanisms by which instrumental music training is related to the implicit processing of emotional prosody remain unclear and merit further exploration.

## Conclusion

Despite the limitations outlined above, our findings do not support an association between instrumental music training and enhanced recognition of emotional prosody in tonal languages. However, they do suggest a link between music training and participants' tendency to assign higher intensity ratings to emotional prosody. This dissociation between emotion recognition and emotion intensity evaluation underscores the complexity of the relationship between music training and emotion perception in tonal languages.

The present study is the first to investigate the relationship between music training and emotional prosody perception in Mandarin, a representative of tonal language. Future research should examine this question using different tonal languages to test the reliability of our results and deepen the understanding of the relationship. Such research has important implications for educators and clinicians, as it may provide a new method to treat impaired implicit processing of emotional prosody.

## Supporting information

**S1 Table. Overview of selected study characteristics for musically trained subjects.** (XLSX)

**S2 Table. Pairwise comparisons of emotional prosodies.** (DOCX)

**S3 Table. Pairwise comparisons of emotional prosodies.** All statistics were obtained from the aligned-and-ranked data. (DOCX)

## Acknowledgments

We are grateful to Dr. Pan Liu from the Department of Psychology at University of Alberta for providing us with the necessary experimental materials.

## Author Contributions

**Conceptualization:** Jun Jiang.

**Data curation:** Mengting Liu.

**Formal analysis:** Jun Jiang.

**Funding acquisition:** Mengting Liu.

**Investigation:** Mengting Liu.

**Methodology:** Mengting Liu, Xiangbin Teng, Jun Jiang.

**Project administration:** Jun Jiang.

**Resources:** Mengting Liu, Jun Jiang.

**Software:** Jun Jiang.

**Supervision:** Jun Jiang.

**Validation:** Jun Jiang.

**Visualization:** Jun Jiang.

**Writing – original draft:** Mengting Liu, Xiangbin Teng, Jun Jiang.

**Writing – review & editing:** Mengting Liu, Xiangbin Teng, Jun Jiang.

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
