## [Decision Letter · Decision Letter 0]

2 Jun 2024

PONE-D-24-10847Impact of instrumental music training on emotional prosody recognition and intensity evaluation in Mandarin speakersPLOS ONE

Dear Dr. Jiang,

Thank you for submitting your manuscript to PLOS ONE. After careful consideration, we feel that it has merit but does not fully meet PLOS ONE’s publication criteria as it currently stands. Therefore, we invite you to submit a revised version of the manuscript that addresses the points raised during the review process.

We look forward to receiving your revised manuscript.

Kind regards,

Junchen Shang

Academic Editor

PLOS ONE

Reviewers' comments:

Reviewer's Responses to Questions

**Comments to the Author**

1. Is the manuscript technically sound, and do the data support the conclusions?

Reviewer #1: Partly

Reviewer #2: Yes

2. Has the statistical analysis been performed appropriately and rigorously? 

Reviewer #1: Yes

Reviewer #2: Yes

3. Have the authors made all data underlying the findings in their manuscript fully available?

Reviewer #1: Yes

Reviewer #2: Yes

4. Is the manuscript presented in an intelligible fashion and written in standard English?

Reviewer #1: Yes

Reviewer #2: Yes

5. Review Comments to the Author

Reviewer #1: There is a lot to like about this study: the research question is well motivated, the sample size is appropriate, the two groups of participants are carefully matched on demographic and cognitive variables, the methods are sound, and the analyses are carefully conducted based on both frequentist and Bayesian statistics. I also think that it is important to publish these null results, particularly considering previous mixed findings obtained with speakers of nontonal languages.

My main concern is with the causal framework adopted by the authors, seen throughout the manuscript from the title (‘Impact of instrumental music training’) to the last paragraph of the discussion. Although they acknowledge briefly that their design precludes inferences of causation, their entire reasoning suggests otherwise. I think that the focus should be on what the study is about – an association – and that the issue of causation needs to be discussed in more detail and critically. Recent reviews of the literature, both focused on emotion recognition specifically (Martins et al., 2021, Emotion Review; Nussbaum & Schweinberger, 2021, Emotion Review), and on music training effects more broadly (Schellenberg & Lima, 2024, Annual Review of Psychology), are directly relevant to this issue.

Other points:

- In the Introduction, the assertion that emotional prosody recognition is important for social interactions would benefit from direct supporting evidence, instead of the indirect evidence that the authors cite (e.g., Neves et al., 2021, Royal Society Open Science).

- On p. 4, the sentence starting on line 71 lacks an ending.

- On p. 5, line 81, the authors say that ‘experimental studies generally support the enhancement of emotional prosody recognition through mixed music training’. The studies that they cite do not provide evidence for such effect, though. This needs to be clarified.

- Why were the stimuli produced by male and female speakers presented in separate blocks?

- Although I understand the rationale for using a composite metric that incorporates speed and accuracy, for comparability with previous studies it would be useful to also present analyses (and descriptives) just focused on accuracy.

- Considering that the novel aspect of this study is the focus on a tonal language, I missed a detailed discussion of how/why differences in linguistic experience could moderate music training effects in prosody recognition. The authors allude to this issue only superficially, without specifying potential reasons for a pattern of null results in tonal languages and predominantly positive results in nontonal ones.

Reviewer #2: This study is interesting. Despite many previous studies on the relationship between language and music, much debate as to what role such relationships play in daily life. This study approached this unclear issue from the perspectives of the characteristics of language.

On the other hand, several points may be difficult for potential readers to understand, so please consider the following points.

p.5, l.96-101 “Recognizing that lexical tones are a common element in both language and music [54] and acknowledging the disparities in acoustic cues employed by listeners of nontonal and tonal languages when interpreting lexical tones [55, 56], it remains unclear whether music training aids in the perception of emotional prosody among native speakers of tonal languages. To fill this gap in current research, the present study aims to explore the connection between music training and the recognition of emotional prosody in tonal languages.”

Comment 1: Given that language differences are one of the main appeal points of this study, the aim of the current study can include international comparisons. If cultural contrasts are included in the scope of the study, the research would be more extensive.

p.17, l.362-371

Comment 2: To shed light on differences of the paper from other studies, the first paragraph of the Discussion should provide a sufficient summary of the appeal points of this study, such as ‘statistical methods’ and ‘large sample size’.

In addition, the appeal points summarized here should also be reflected in the abstract.

p.18, l.366 “Interestingly, music majors assigned higher intensity ratings to happy, fearful, and angry prosodies.”

Comment 3: Why were differences observed in these three intensities? It would be important to elucidate this cause within Discussion for explanations of the overall results.

p.18, l.380-382 “This inconsistency may be explained by the intricate interplay between music and language experience in shaping emotional prosody recognition”

Comment 4: In several studies of absolute pitch, the high number of absolute pitch holders in the Asian region is sometimes explained by linguistic characteristics. These findings of absolute pitch may also provide clues for interpreting the present results.

p.19, l.393-394 “The discrepancy may be attributed to the difference in sample sizes;”

Comment 5: Are these differences really only due to sample size? The differences due to language can be possible. If so, why do they differ by language? These factors should be discussed more carefully. This would deepen cross-cultural aspects in this research topic.

p.20, l.412-414

“Thus, our findings suggest that instrumental music training may primarily influence the implicit, rather than explicit, processing of emotional prosody in tonal languages.”

Comment 6: Why were these results obtained and what processes in people can be attributed to these results? More adequate discussions here should be useful to deepen author(s)’ findings.

p.20, l.417-418

“Future studies should employ true experimental designs to address this issue.”

Comment 7: What is “true” experimental design? And how can it be achieved?

p.20, l.420-421

“Third, this investigation utilized only one tonal language, raising questions about the generalizability of the findings to other tonal languages.”

Comment 8: If there was a table, which could list the previous studies, it would be easier for the reader to understand the novelty of this study. For example, for each previous study, the language(s) covered, the type of musical experience and the age of the participants.

p.20, l.425-329

Comment 9: This last part of discussion can be separated Conclusion section. A succinct but sufficient summary of importance of this research, the development of future research (e.g. contrasts with Chinese and languages close to it are probably yet to be done), and the applications can provide stronger impressions to potential readers.

In particular, how this research can be applied in real-life situations, social issues or human evolution?

6. PLOS authors have the option to publish the peer review history of their article (what does this mean?). If published, this will include your full peer review and any attached files.

Reviewer #1: No

Reviewer #2: No

---

## [Author Response · Author response to Decision Letter 0]

11 Jul 2024

We have carefully considered the reviewers' comments and uploaded them into the system as a separate document.

---

## [Decision Letter · Decision Letter 1]

13 Aug 2024

Instrumental music training relates to intensity assessment but not emotional prosody recognition in Mandarin

PONE-D-24-10847R1

Dear Dr. Jiang,

We’re pleased to inform you that your manuscript has been judged scientifically suitable for publication and will be formally accepted for publication once it meets all outstanding technical requirements.

Kind regards,

Junchen Shang

Academic Editor

PLOS ONE

Additional Editor Comments (optional):

Reviewers' comments:

Reviewer's Responses to Questions

**Comments to the Author**

1. If the authors have adequately addressed your comments raised in a previous round of review and you feel that this manuscript is now acceptable for publication, you may indicate that here to bypass the “Comments to the Author” section, enter your conflict of interest statement in the “Confidential to Editor” section, and submit your "Accept" recommendation.

Reviewer #1: All comments have been addressed

Reviewer #2: (No Response)

2. Is the manuscript technically sound, and do the data support the conclusions?

Reviewer #1: Yes

Reviewer #2: Yes

3. Has the statistical analysis been performed appropriately and rigorously? 

Reviewer #1: Yes

Reviewer #2: Yes

4. Have the authors made all data underlying the findings in their manuscript fully available?

Reviewer #1: Yes

Reviewer #2: Yes

5. Is the manuscript presented in an intelligible fashion and written in standard English?

Reviewer #1: Yes

Reviewer #2: Yes

6. Review Comments to the Author

Reviewer #1: (No Response)

Reviewer #2: (No Response)

7. PLOS authors have the option to publish the peer review history of their article (what does this mean?). If published, this will include your full peer review and any attached files.

Reviewer #1: No

Reviewer #2: No

---

## [Editor Report · Acceptance letter]

21 Aug 2024

PONE-D-24-10847R1 

PLOS ONE

Dear Dr. Jiang, 

I'm pleased to inform you that your manuscript has been deemed suitable for publication in PLOS ONE. Congratulations! Your manuscript is now being handed over to our production team.

Kind regards, 

on behalf of

Dr. Junchen Shang 

Academic Editor

PLOS ONE